# Morphological and Morphometric Analysis of Canine Choroidal Layers Using Spectral Domain Optical Coherence Tomography

**DOI:** 10.3390/ijerph20043121

**Published:** 2023-02-10

**Authors:** Jowita Zwolska, Ireneusz Balicki, Agnieszka Balicka

**Affiliations:** 1Department and Clinic of Animal Surgery, Faculty of Veterinary Medicine, University of Life Sciences in Lublin, 20-950 Lublin, Poland; 2Small Animals Clinic, University of Veterinary Medicine and Pharmacy in Kosice, 041 81 Kosice, Slovakia

**Keywords:** canine, choroidal layers, morphometry, optical coherence tomography, uvea

## Abstract

The choroid, a multifunctional tissue, has been the focus of research interest for many scientists. Its morphology and morphometry facilitate an understanding of pathological processes within both the choroid and retina. This study aimed to determine the choroidal layer thicknesses in healthy, mixed-breed mesocephalic dogs, both male (M) and female (F), using spectral domain optical coherence tomography (SD-OCT) with radial, cross-sectional, and linear scans. The dogs were divided into two groups based on age: middle-aged (MA) and senior (SN). Thicknesses of choroidal layers, namely RPE–Bruch’s membrane–choriocapillaris complex (RPE-BmCc) with tapetum lucidum in the tapetal fundus, the medium-sized vessel layer (MSVL), and the large vessel layer with lamina suprachoroidea (LVLS), as well as whole choroidal thickness (WCT), were measured manually using the caliper function integrated into the OCT software. Measurement was performed dorsally and ventrally at a distance of 5000–6000 μm temporally and nasally at a distance of 4000–7000 μm to the optic disc on enhanced depth scans. The measurements were conducted temporally and nasally in both the tapetal (temporal tapetal: TempT, nasal tapetal: NasT) and nontapetal (temporal nontapetal: TempNT, nasal nontapetal: NasNT) fundus. The ratio of the MSVL thickness to the LVLS thickness for each region was calculated. In all examined dogs, the RPE-BmCc in the dorsal (D) region and MSVL in the Tt region were significantly thicker than those in the other regions. The MSVL was thinner in the ventral (V) region than in the D, TempT, TempNT and NasT regions. The MSVL was significantly thinner in the NasNT region than in the D region. LVLS thickness and WCT were significantly greater in the D and TempT regions than those in the other regions and significantly lesser in the V region than those in the other regions. The MSVL-to-LVLS thickness ratio did not differ between the age groups. Our results reveal that the choroidal thickness profile does not depend on age. Our findings can be used to document the emergence and development of various choroidal diseases in dogs in the future.

## 1. Introduction

The choroid, a multifunctional tissue, is responsible for maintaining retinal temperature, supplying its outer layers with nutrients and oxygen, and playing a role in intraocular pressure (IOP) modulation [1]. Studies also point to its role in ocular focus normalization, including emmetropization, and thereby the regulation of eye growth [2,3].

Additionally, the choroid expresses and synthesizes a variety of signal molecules, several matrix metalloproteases and their tissue inhibitors [4], as well as growth factors that are involved in angiogenesis [5]. One of the first studies to provide evidence of the impact of choroidal cell growth factor on other tissues demonstrated that explants of the choroid target tissues are essential for the development and survival of ciliary ganglion neurons in culture systems [6]. Profound knowledge of normal choroidal morphology and morphometry provides the ability to understand pathological processes within both the choroid and retina.

The choroid is an anatomical structure located between the retinal pigment epithelium and the sclera. It is primarily composed of blood vessels, nerves, and pigment cells. Bruch’s membrane, which is adjacent to the retina, adheres to the external layer of capillaries, which constitute the choriocapillaris. Histologically, the choroid has been divided into four or six layers, depending on whether the vascular region is considered to be one- or two-layered and whether the lamina fusca is considered to be of scleral or choroidal origin. In most domestic mammals, starting from the retinal (inner) side, the choroid consists of the Bruch’s membrane, choriocapillaris, tapetum lucidum, intermediate-vessel layer, large-vessel layer, and suprachoroidea [7].

Spectral-domain optical coherence tomography (SD-OCT) is a non-invasive and non-contact method that provides rapid, cross-sectional, and real-time in vivo imaging of the retina and choroid [8]. This technique is also used as an objective method for measuring and analyzing each retinal layer [9,10,11,12].

The choroidal vascular system has been the focus of research interest for many scientists. Until the invention of optical coherence tomography, studies of the choroid were limited to post-mortem examinations. The latest technologies allow non-invasive imaging, along with the qualitative and quantitative in vivo assessment of the human choroid [13]. At present, most ophthalmologists manually define the boundaries of the choroid to measure its thickness; however, some studies have introduced different automatic choroidal segmentation methods [14,15,16]. The choroidal vasculature has been broadly investigated in humans. A correlation between choroidal dimensions and patient characteristics, such as age, refractive error, corneal refractive power, ethnicity, body height, and sexual maturation, has been observed in numerous studies involving healthy human subjects [17,18,19,20]. Many studies have investigated choroidal profiles in humans [21,22,23]. The morphological features of the choroid in both physiological and pathological conditions indicate that this vascular structure plays a crucial role in many chorioretinal disorders [24]. OCT has become increasingly essential for the diagnosis of many ocular and systemic diseases. According to Mano et al., in the case of ocular amyloidosis, the evaluation of choroidal characteristics may serve as a biomarker for systemic involvement [25].

Few studies have documented the use of OCT to measure the choroid in guinea pigs [26,27,28,29,30]. Li et al. used OCT to assess the retinal and choroidal thickness in guinea pigs for comparison with the histological sections. The authors reported good agreement between the in vivo and histological measurements [29]. Nonetheless, there has been no research concerning the in vivo analysis of canine choroidal layers.

In our previous study, we analyzed whole choroidal thickness in dogs in the dorsal (D), ventral (V), temporal (T), and nasal (N) regions [31]. Choroidal thickness in dogs depends on the area being assessed and not on the age of the dogs. No statistically significant difference was observed in the whole choroidal thickness in each choroidal region between dogs of different age groups. In both senior (SN) and middle-aged (MA) mixed-breed dogs, the D region was the thickest, followed by the T region, and the V region was the thinnest [31]; however, in the previous study, division into choroidal layers was not considered. 

The purpose of this study was to perform morphological and morphometric in vivo analyses of the choroidal profiles in MA and SN dogs.

## 2. Materials and Methods

### 2.1. Animals and Procedures

Forty-five clinically healthy mixed-breed mesocephalic dogs, both male (M) and female (F), were examined in this study. The dogs were divided into two groups based on their age: MA and SN. The classification was based on a dog age chart indicating the relationship between age and weight [32]. Dogs in the MA group (*n* = 21; 9 M and 12 F) were 4–7 years old and weighed 12–34 kg. The dogs in the SN group (*n* = 24; 12 M and 12 F) were 8–13 years old and weighed 13–32 kg. All the males involved in the study were unneutered. The spay status of 5 females was unknown, while the remaining 19 females were non-spayed.

The animals were patients at the Department and Clinic of Animal Surgery at the University of Life Sciences in Lublin. The dog owners were informed about the details of the clinical trials, and their consent was obtained. The study was performed in accordance with the National Research Council’s Guide for the Care and Use of Laboratory Animals, Polish law and with Directive 2010/63/EU of the European Parliament and of the Council of 22 September, 2010, on the protection of animals used for scientific purposes, Chapter I, Article 1, point 5(b). The study was also approved by the Scientific Research Committee of the Department and Clinic of Animal Surgery at the University of Life Sciences in Lublin (#3/2018) concerning non-experimental clinical patients. Some of the patients in whom measurements of individual choroidal layers were performed were patients included in the authors’ most recent studies on the total thickness of the choroid [31].

All dogs were classified as healthy based on the physical examination, including the cardiological examination (echocardiographic, arterial blood pressure), blood tests, and comprehensive ophthalmological examinations. Blood tests included complete blood cell count and measurement of urea, complete bilirubin, creatinine, aspartate transaminase, alanine transaminase, alkaline phosphatase, and amylase levels. The dogs were dewormed twice a year. Patients with a history of any retinal disease in either eye were excluded.

Ocular examinations were performed using a slit-lamp biomicroscope (Shin-Nippon, Tokyo, Japan). Fundus examinations were performed using a binocular indirect ophthalmoscope (Keeler, Windsor, UK), direct ophthalmoscope (Welch Allyn, Chicago, IL, USA), and panoptic ophthalmoscope (Welch Allyn, Chicago, IL, USA). Photographs of the ocular fundus were taken using a Handy NM-200D fundus camera (Nidek, Gamagori, Japan) connected to a computer operating the IrfanView software (Wiener Neustadt, Austria). In all dogs, the pupillary light reflex (both direct and consensual), dazzle reflex, menace response, and palpebral reflex were estimated. Behavioral ophthalmic tests, including tracking and placing, were conducted along with obstacle tests under scotopic and photopic conditions. The IOP was measured using a rebound tonometer (TonoVet, iCare, Vantaa, Finland) and was found to be 15–20 mmHg. The Schirmer’s tear test I (Eickemeyer, Tuttlingen, Germany) was performed bilaterally in all patients. No vision impairment or ocular abnormalities were identified in any dog. Animals with any ocular abnormalities were excluded from the study, e.g., retinal degeneration, retinal dysplasia, sudden acquired retinal degeneration syndrome (SARDS), retinal detachment, and retinal hemorrhage. 

Pupils were dilated with tropicamide eye drops (Tropicamidum WZF 1%; Polfa Warszawa S.A., Warsaw, Poland). Sedation was carried out using the intramuscular administration of medetomidine (0.03 mg/kg; Cepetor 1 mg/mL, CP Pharma, Burgdorf, Germany), and local anesthesia of the corneal and conjunctival surface was achieved using 0.5% proxymetacaine (Alcaine 5 mg/mL, Alcon, Warsaw, Poland). A timeframe was set for the OCT examination from 9:00 a.m. to 1:00 p.m. The examination was performed 15–30 min after medetomidine administration. Thumb forceps were used to grasp the bulbar conjunctiva and stabilize the eyes. During the SD-OCT examination, the cornea was moistened every 30 s with saline. 

### 2.2. OCT Scan and Data Analysis

Imaging was performed for both eyes of each animal using SD-OCT (wavelength: 840 nm; scan pattern: enhanced depth imaging, EDI; Topcon 3D OCT-2000, Topcon, Japan) with linear, cross-sectional, and 6-line radial scans. The device software allowed the determination of the precise location of choroidal measurements on the resulting scans. The caliper function integrated into the OCT software was used to collect manual measurements of the choroid.

The thickness of the choroidal layers, namely RPE–Bruch’s membrane–choriocapillaris complex (RPE-BmCc) with tapetum lucidum in the tapetal fundus, the medium-sized vessel layer (MSVL), the large vessel layer with lamina suprachoroidea (LVLS), and the whole choroidal thickness (WCT; Figure 1) were measured manually using the caliper function integrated into the OCT software. RPE-BmCc was defined as a layer with a spotted appearance [33] and a highly reflective border in the tapetal regions.

During certain scans, the choriocapillaris was identifiable (Figure 2), but in many cases, it was difficult or impossible to separate the capillary from the retinal pigment epithelium, Bruch’s membrane, and tapetum lucidum. Therefore, these structures were assessed as a single layer—RPE-BmCc. The appearance of the RPE-BmCc was more reflective than MSVL.

The MSVL is defined as a layer containing medium-sized choroidal vessels measuring ≤100 μm, which are visualized as medium-sized, hypo-intense spaces surrounded by a hyper-intensive stroma. The MSVL was located immediately beneath the RPE-BmCc. The LVLS is classified as an outer choroidal layer containing large choroidal vessels measuring ≥100 μm [34]. The outer boundary of the LVLS is defined as the choroidal scleral interface on the SD-OCT images. The major criteria to differentiate between the MSVL and LVLS layers are the choroidal vessel lumen and the density of the inter-vascular tissue [21]. We used the method suggested by Branchini et al. to establish the outer borders of the choroid and choroid-sclera junction, which was described as a hyperreflective line and was localised externally to the large choroidal vessels [34]. The choroidal thickness was defined as the vertical distance from the hyperreflective line of the RPE–Bruch’s membrane complex to the hyperreflective line of the inner surface of the sclera [34,35]. The measurements were performed dorsally and ventrally at a distance of 5000–6000 μm to the optic disc. Temporal (T) and nasal (N) scans were taken at a distance of 4000–7000 μm to the optic disc (Figure 3 and Figure 4). The measurements were obtained on enhanced depth scans. The measurements were conducted temporally and nasally in both the tapetal (temporal tapetal: TempT, nasal tapetal: NasT) and nontapetal (temporal nontapetal: TempNT, nasal nontapetal: NasNT) fundus. Measurements in the TempT and NasT regions were taken at a distance of 500–2000 µm dorsally from the border between the tapetal and nontapetal regions. Measurements in the temporal and nasal nontapetal regions were taken at a distance of 500–2000 µm ventrally from the border between the tapetal and nontapetal regions. Three measurements were performed for each region: the first one was in the center of the scan, and the other two were on the right and left at a distance of 1500 μm from the center. The average of the three measurements for each segment was calculated. In order to avoid variability, all measurements were made at one time and approved by the same two authors.

### 2.3. Statistical Methods

The database and statistical analyses were carried out using Statistica 9.1 software (StatSoft, Cracow, Poland). Quantitative data are presented as mean  ±  standard deviation (SD). The normality of the distribution of variables in the studied groups was checked using the Shapiro–Wilk normality test. Student’s t-test was used to test the differences between the MA and SN groups. The differences in layer thickness and whole choroidal thickness between regions were assessed using analysis of variance (ANOVA). Tukey’s honestly significant difference post hoc test was performed if an overall significance was observed. Statistical significance was set at *p* < 0.05.

## 3. Results

The mean thicknesses of the choroidal layers in the D, V, TempT, TempNT, NasT, and NasNT regions are presented in Table 1.

No significant age-related differences were observed in the thicknesses of the respective choroidal regions. Owing to the absence of age-related dissimilarities, the differences in layer and whole choroidal thickness between regions were assessed without dividing the dogs into different age groups, and the measurements were performed on 90 eyes of 45 dogs (Table 2). In all the examined dogs (*n* = 45; 21 M and 24 F), the RPE-BmCc in the D region (*p* < 0.001) and MSVL in the TempT region (*p* < 0.005) were significantly thickest compared to those in the other regions. In addition, RPE-BmCc was significantly thicker in the TempT region than in the V region (*p* < 0.05). Moreover, MSVL was thinner in the V region than that in the D (*p* < 0.001), TempT (*p* < 0.001), TempNT (*p* < 0.05), and NasT (*p* < 0.001) regions. The MSVL was significantly thinner in the nasal nontapetal region than in the D region (*p* < 0.05). LVLS thickness and WCT were significantly greater in the D and TempT regions than in the other regions (*p* < 0.001) and significantly lesser in the V region than in the other regions (*p* < 0.005) (Table 2).

The MSVL-to-LVLS thickness ratio was not age-related in any of the canine choroidal regions (Table 3). This ratio was similar for each choroidal region, amounting to 0.28 ± 0.26 D, 0.32 ± 0.33 V, 0.32 ± 0.37 in the TempT region, 0.32 ± 0.32 in the TempNT region, 0.35 ± 0.34 in the NasT region, and 0.29 ± 0.33 in the NasNT region in the MA and SN groups, respectively.

## 4. Discussion

Studies in which EDI with SD-OCT has been used have presented certain methods of performing manual measurements of human choroidal vascular layers. The scans obtained during our research allowed the precise determination of the individual layers of the choroid. The method suggested by Branchini et al. [34] for analyzing vessel lumen diameter has been frequently applied in various studies [22,36,37]. According to previous studies, MSVL is defined as the layer containing medium-sized choroidal vessels measuring ≤100 μm, and LVLS is defined as the outer choroidal layer containing large choroidal vessels measuring ≥100 μm [33]. In our study, the location of the RPE–Bruch’s membrane complex and the inner scleral surface was determined using a commonly used human ophthalmological method [37]. Moreover, previous work employing SD-OCT in the field of veterinary ophthalmology for retinal imaging was referred to [38,39,40,41,42,43]. We determined the outer border of the choroid (i.e., the inner surface of the sclera) based on the veterinary ophthalmology literature describing the histological structure of the dog choroid [44] and human ophthalmology literature describing the method of measuring the thickness of choroid [45,46,47,48,49,50,51].

Choriocapillaris is a highly anastomosed capillary network located externally from Bruch’s membrane. The choriocapillaris arises from the arterioles in the medium-sized vessel layer, also known as Sattler’s layer [1]. In humans, the choriocapillaris layer is approximately 10 μm thick at the fovea, where the density of capillaries is the highest, thinning up to approximately 7 μm at the fundus periphery [1]. In our study, the innermost choroidal layer, which consisted of the RPE-BmCc and tapetum lucidum in the tapetal fundus, was thickest in the D region (27.42 µm). In the other choroidal regions, it was considerably thinner and thinnest in the nontapetal regions, ventrally declining up to 7.99 µm.

Yamaue et al. (2014) conducted a histological examination of the canine choroid. The tapetum is described as a rounded triangle with a smooth contour. The base is usually in contact with the optic nerve disc. The study reported that the thickest part of the tapetum is located dorso-temporally to the optic disk, varying in thickness from 20 to 70 µm, decreasing with increasing distance from the center of the tapetum [52]. According to Lesiuk and Breakevelt, the central part of the canine tapetum lucidum contains light-reflecting material and consists of 15–20 layers of polygonal, zinc-rich cells with a mean thickness of 3–28 µm. The number of cell layers decreases with increasing distance from the center of the tapetum. Bruch’s membrane also varies in thickness and is thinnest in the areas containing the tapetum lucidum [53]. In our study, the first choroidal layer, which consisted of the RPE-BmCc and tapetum lucidum, was the thickest in the tapetal fundus, dorsally amounting to 27.42 µm, decreasing up to 10.81 µm in the Tt and 9.56 µm in the Nt regions. However, these values correspond to the thickness of the whole RPE–Bruch’s membrane–choriocapillaris–tapetum lucidum complex.

A previous study identified the choriocapillaris in four animal species that lack a tapetum lucidum. The choriocapillaris was visualized as a hyporeflective layer between the RPE–Bruch’s membrane complex and the outer choroid [54]. Mischi et al. evaluated choriocapillaris visibility imaged with SD-OCT in dogs and cats. According to the study, the choriocapillaris in dogs was visible in over 80% and over 90% of scanlines, depending on the grader [55]. The present study shows that the reflective nature of the tapetum lucidum interferes with the visibility of the choriocapillaris and connecting vasculature. The choriocapillaris could be differentiated on certain SD-OCT scans; however, it was often difficult or impossible to separate the capillary network from the RPE–Bruch’s membrane complex and tapetum lucidum for proper measurement (Figure 2). Therefore, we presented it as an RPE-BmCc-tapetum lucidum complex.

One of the aims of this study was to assess possible age-related variations in the thickness of different choroidal layers in MA and SN dogs. Ruiz-Medrano et al. studied the choroidal profiles of different age groups in a healthy human population. The mean subfoveal choroidal and mean horizontal Haller’s layer thickness in the different age groups decreased progressively with increasing age. The mean subfoveal choriocapillaris and Sattler’s thickness did not exhibit this trend toward progressive thinning with increasing age in humans [56]. Kamal Abdellatif et al. studied 125 healthy Egyptians aged between 20 and 79 years. The retinal pigment epithelium with photoreceptor outer segment thickness showed significant thinning with increasing age and decreasing choroidal thickness. Both the retinal pigment epithelium with an outer photoreceptor segment and choroidal thickness showed a statistically significant negative correlation with age [57]. Our results did not reveal any age-related differences in the choroidal profiles of the dogs. No significant differences were observed in the thicknesses of the RPE-BmCc, MSVL, LVLS and WCT in each choroidal region between the MA and SN groups.

Research conducted by Xu et al. showed that in humans, a higher peripapillary small-to-medium vessel layer (SMVL)-to-peripapillary large vessel layer (LVLS) thickness ratio was related to older age, suggesting a preferential age-related thinning of the LVLS in humans. The study revealed that the SMVL-to-LVLS thickness ratio is significantly the highest in the temporal region and the lowest in the superonasal, superior and nasal regions [23]. Our study showed that the MSVL-to-LVLS thickness ratio was not age-related in dogs. Comparing the results of the MA and SN groups, the ratio values were similar for each choroidal region.

In our previous study, we analyzed whole choroidal thickness in dogs in the D, V, T, and N regions. However, the T and N regions were not differentiated into tapetal and nontapetal regions [31]. Choroidal thickness in the D and V regions was comparable to the values reported in the present study. The choroidal thickness in the temporal region was found to be 152 and 151 µm in the MA and SN groups, respectively. Moreover, the choroidal thickness in the N region was found to be 135 and 132 µm in the MA and SN groups, respectively [31]. In the present study, the choroidal thickness was 175 and 182 µm in the TempT region in the MA and SN groups, respectively. Furthermore, the choroidal thickness was 139 and 144 µm in the NasT regions in the MA and SN groups, respectively. The mean TempNT choroidal thickness was 119 and 138 µm in the MA and SN groups, respectively. Furthermore, the mean NasNT choroidal thickness was 124 and 128 µm in the MA and SN groups, respectively. The present study indicates that the choroidal thickness varies in the N and T regions, depending on whether the fundus is tapetal or nontapetal. The choroidal thickness in both T and N regions in the tapetal fundus was much greater than in the nontapetal fundus. Interestingly, this was not related to the presence of the tapetum lucidum. The thickness of the tapetum lucidum decreases in the peripheral areas [52]. This phenomenon is consistent with our findings, where the mean thickness of RPE-BmCc was 27.42 µm in the D, 10.81 µm in the TempT, and 9.56 µm in the Nt regions. The reason for this considerable difference in the thickness of the choroid in the N and T regions between the tapetal and nontapetal groups was the difference in the thickness of the LVLS layer. The average thickness of the LVLS layer was 126.14 µm in the TempT, 94.96 µm in the TempNT, 99.81 µm in the NasT, and 90.15 µm in the NasNT regions. Our study showed that the choroidal vascular layers, MSVL and LVLS, were the thickest in the TempT region. A study performed by Mowat et al. in 9 beagle dogs showed that the area centralis is likely to be centered at a point 1.5 ± 0.2 mm temporal and 0.6 ± 0.1 mm superior to the optic disc, within the tapetal fundus [58]. We conclude that the area centralis might be associated with a higher choroidal blood supply. This needs to be further studied. However, the TempT region was not found to be associated with the thickest RPE-BmCc–tapetum lucidum layer, which is consistent with the previous studies. In histological sections of the retina and choroid, performed by Yamaue et al., the visual streak was located within the tapetal region but not on the thickest part of the tapetum. The tapetum is well suited for night vision. Therefore, it would impair visual resolution in daylight vision by scattered light [59].

Choroidal thickness demonstrates diurnal variation in antiphase with axial elongation in animal models and humans [2,60,61]. In chickens, choroidal thickness is the highest at midnight and starts to decrease later during the night [2]. Similarly, the choroidal thickness increases in humans over the course of the day. In this study, all measurements were obtained between 9 a.m. and 1 p.m. to minimize the potential for diurnal variations in choroidal thickness. This timeframe was determined based on the reports demonstrating that changes in choroidal thickness were minimal during those hours and varied by no more than 3 μm [61].

We employed the same sedation protocol in all dogs to eliminate the possibility of anesthesia-induced variation in choroidal thickness. Research on the sedative and cardiorespiratory effects of intramuscular administration of medetomidine revealed that the increase in the mean arterial blood pressure in dogs occurs up to 10 min after the administration and then remains at a constant level until atipamezole administration [62]. Therefore, SD-OCT was performed between 15 and 30 min after medetomidine administration.

The present study provides the first Spectralis SD-OCT-based data on the vascular layers of the choroid in dogs. The aim of this study was to determine the choroidal thickness profile in SN and MA mixed-breed mesocephalic dogs, both M and F. Our aim was not to narrow our research to a particular dog breed. Many ocular diseases are diagnosed in mixed breeds; for example, SARDS is a disease often diagnosed in mixed-breed dogs [63]. This group of patients could serve as a reference while studying other dog species in the future. Our findings can also be used as an indicator for purebred dogs and to study and compare the choroidal vascular layer measurements under different disease conditions in certain dog breeds, such as SARDS, progressive retinal atrophy, and local retinopathy.

Knowledge of the characteristics of individual choroidal vascular layers and their responses to certain disease conditions is limited in veterinary medicine. The assessment of the choroid in clinically healthy animals is required for further research to elucidate the correlation between the occurrence of each disease and vascular alterations. Additionally, the limitation of our research is the possible influence of choroidal blood perfusion on the choroidal thickness. This will require further research. Other limitations are the manual caliper. In order to avoid inter-observer variability, all measurements were made at one time and accepted by the same two authors. Measurements made with the manual calipers are not as accurate as with the automated analysis of the OCT scans.

## 5. Conclusions

The present study provides the first Spectralis SD-OCT-based data on choroidal vascular layer thickness in dogs. Moreover, our results revealed that the choroidal thickness profile does not depend on age. Furthermore, the MSVL-to-LVLS thickness ratio in dogs was not found to be age-related. Our findings can be used to study and compare whole choroidal and vascular layer thickness measurements in different disease conditions in different populations. SD-OCT imagery of the posterior eye segment can be used to document the emergence and development of various choroidal diseases in dogs in the future. The findings and present group of patients may be a reference point for further studies on purebred dogs.

## Figures and Tables

**Figure 1 ijerph-20-03121-f001:**
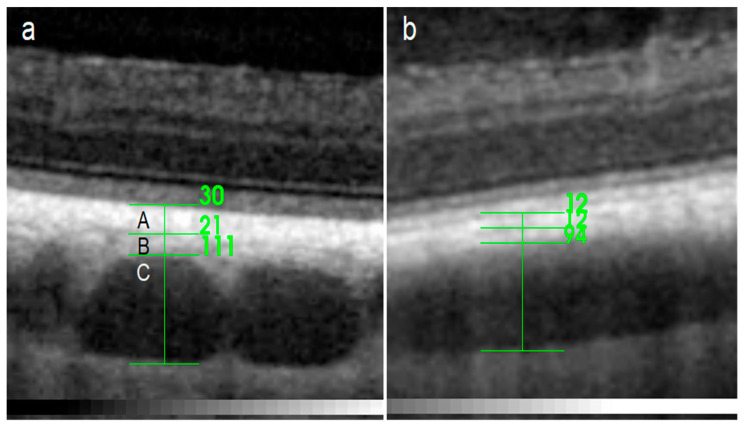
Spectral-domain optical coherence tomography (SD-OCT) scan with the choroidal layer measurements in the tapetal (**a**) and nontapetal (**b**) fundus: (A) RPE–Bruch’s membrane–choriocapillaris complex (RPE-BmCc) with tapetum lucidum, (B) medium-sized vessel layer (MSVL), and (C) large vessel layer with lamina suprachoroidea (LVLS).

**Figure 2 ijerph-20-03121-f002:**
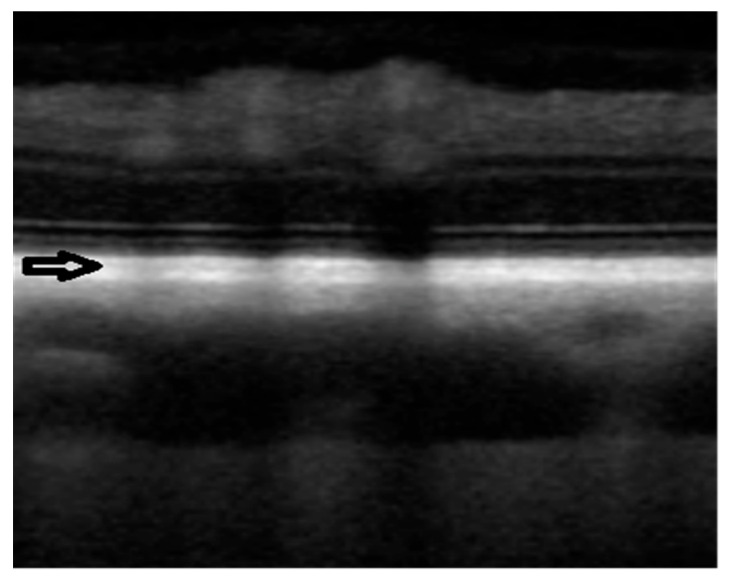
SD-OCT scan taken in the dorsal tapetal fundus. The black arrow indicates choriocapillaris (hyporeflective line). The choriocapillaris is located between the RPE–Bruch’s membrane complex and the tapetum lucidum.

**Figure 3 ijerph-20-03121-f003:**
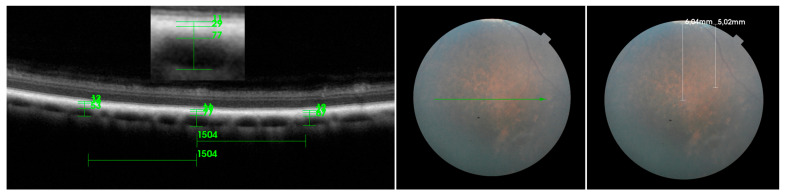
Measurements of choroidal thickness in the ventral region on SD-OCT scan (green arrow in the fundus photograph). Three measurements were conducted for each analyzed segment: the first one in the center of the scan and the other two on the right and left at a distance of 1500 µm from the center. The measurements were performed at a distance of 5000–6000 µm to the optic disc (white arrows in the fundus photograph).

**Figure 4 ijerph-20-03121-f004:**
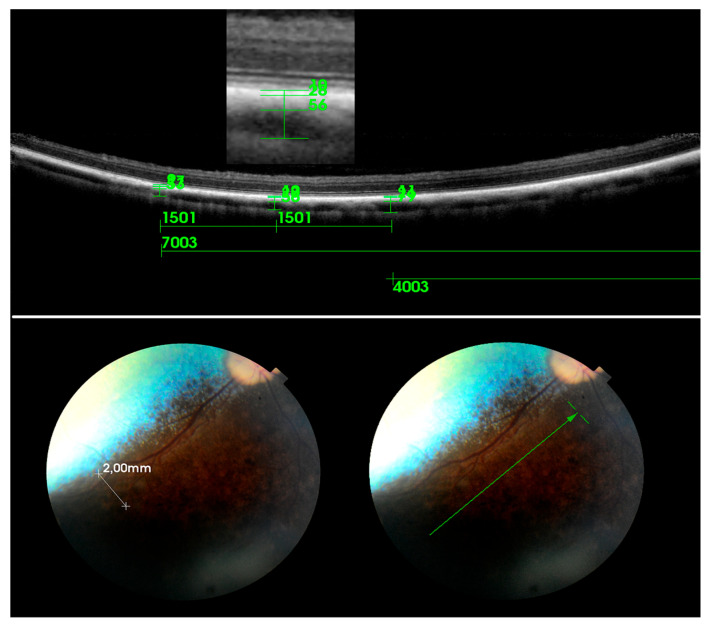
Measurements of choroidal layer thickness in the nasal nontapetal region on SD-OCT scan (green arrow in the fundus photograph). Measurements on linear scans were performed at distances of 4000–7000 µm. Measurements in the temporal and nasal nontapetal regions were taken at a distance of 500–2000 µm ventrally from the border between the tapetal and nontapetal regions (white arrows in the fundus photograph).

**Table 1 ijerph-20-03121-t001:** Mean choroidal layer thickness (RPE–Bruch’s membrane–choriocapillaris complex: RPE-BmCc, medium-sized vessel layer: MSVL, large vessel layer with lamina suprachoroidea: LVLS) and whole choroidal thickness (WCT) (µm) ± SD in dorsal (D), ventral (V), temporal tapetal (TempT) and nontapetal (TempNT), as well as nasal tapetal (NasT) and nontapetal (NasNT), regions in middle-aged (MA) and senior (SN) dogs.

	RPE-BmCc	MSVL	LVLS	WCT
	D			
MA	28.7 ± 7.49	32.59 ± 7.83	121.72 ± 24.58	183.02 ± 30.15
SN	26.44 ± 5.04	31.62 ± 7.62	124.87 ± 28.32	182.93 ± 32.01
	V			
MA	7.71 ± 0.96	24.43 ± 8.07	78.21 ± 19.22	110.24 ± 23.50
SN	8.13 ± 0.50	24.33 ± 6.21	75.39 ± 14.28	107.84 ± 17.68
	TempT			
MA	10.76 ± 2.17	38.15 ± 7.64	126.78 ± 35.78	175.69 ± 38.58
SN	10.90 ± 2.38	45.43 ± 10.29	125.67 ± 19.96	182.00 ± 23.54
	TempNT			
MA	8.12 ± 0.91	27.11 ± 9.87	84.45 ± 17.24	119.68 ± 24.81
SN	8.25 ± 0.68	30.86 ± 6.94	99.59 ± 22.98	138.62 ± 26.81
	NasT			
MA	9.83 ± 1.43	32.90 ± 10.02	96.78 ± 12.56	139.52 ± 15.67
SN	9.33 ± 1.47	33.03 ± 5.40	102.33 ± 24.48	144.69 ± 24.06
	NasNT			
MA	8.05 ± 0.86	26.34 ± 8.90	89.72 ± 14.19	124.12 ± 19.40
SN	8.44 ± 0.69	29.74 ± 7.50	90.59 ± 12.43	128.77 ± 16.89

**Table 2 ijerph-20-03121-t002:** Differences in mean thicknesses (M ± SD) of RPE-BmCc, MSVL, LVLS and WCT in all examined dogs between dorsal (D), ventral (V), temporal tapetal (TempT), temporal nontapetal (TempNT), nasal tapetal (NasT), and nasal nontapetal (NasNT) regions.

**Layer**	**Region**	**Mean ± SD**	**Statistical Significance**	**Layer**	**Region**	**Mean ± SD**	**Statistical Significance**
RPE-BmCc	D vs. V	27.42 ± 6.28 vs. 7.99 ± 0.70	*p* < 0.001	MSVL	D vs. V	27.42 ± 6.28 vs. 7.99 ± 0.70	*p* < 0.001
	D vs. TempT	27.42 ± 6.28 vs. 10.84 ± 2.24	*p* < 0.001		D vs. TempT	27.42 ± 6.28 vs. 10.84 ± 2.24	*p* < 0.001
	D vs. TempNT	27.42 ± 6.28 vs. 8.21 ± 0.75	*p* < 0.001		D vs. TempNT	27.42 ± 6.28 vs. 8.21 ± 0.75	*p* = 0.67
	D vs. NasT	27.42 ± 6.28 vs. 9.56 ± 1.44	*p* < 0.001		D vs. NasT	27.42 ± 6.28 vs. 9.56 ± 1.44	*p* = 1.00
	D vs. NasNT	27.42 ± 6.28 vs. 8.24 ± 0.79	*p* < 0.001		D vs. NasNT	27.42 ± 6.28 vs. 8.24 ± 0.79	*p* < 0.05
	V vs. TempT	7.99 ± 0.70 vs. 10.84 ± 2.24	*p* < 0.05		V vs. TempT	7.99 ± 0.70 vs. 10.84 ± 2.24	*p* < 0.001
	V vs. TempNT	7.99 ± 0.70 vs. 8.21 ± 0.75	*p* = 1.00		V vs. TempNT	7.99 ± 0.70 vs. 8.21 ± 0.75	*p* < 0.05
	V vs. NasT	7.99 ± 0.70 vs. 9.56 ± 1.44	*p* = 0.47		V vs. NasT	7.99 ± 0.70 vs. 9.56 ± 1.44	*p* < 0.001
	V vs. NasNT	7.99 ± 0.70 vs. 8.24 ± 0.79	*p* = 1.00		V vs. NasNT	7.99 ± 0.70 vs. 8.24 ± 0.79	*p* = 0.11
	TempT vs. TempNT	10.84 ± 2.24 vs. 8.21 ± 0.75	*p* = 0.08		TempT vs. TempNT	10.84 ± 2.24 vs. 8.21 ± 0.75	*p* < 0.001
	TempT vs. NasT	10.84 ± 2.24 vs. 9.56 ± 1.44	*p* = 0.85		TempT vs. NasT	10.84 ± 2.24 vs. 9.56 ± 1.44	*p* < 0.005
	TempT+ vs. NasNT	10.84 ± 2.24 vs. 8.24 ± 0.79	*p* = 0.05		TempT vs. NasNT	10.84 ± 2.24 vs. 8.24 ± 0.79	*p* < 0.001
	TempNT vs. NasT	8.21 ± 0.75 vs. 9.56 ± 1.44	*p* = 0.73		TempNT vs. NasT	8.21 ± 0.75 vs. 9.56 ± 1.44	*p* = 0.64
	TempNT vs. NasNT	8.21 ± 0.75 vs. 8.24 ± 0.79	*p* = 1.00		TempNT vs. NasNT	8.21 ± 0.75 vs. 8.24 ± 0.79	*p* = 0.92
	NasT vs. NasNT	9.56 ± 1.44 vs. 8.24 ± 0.79	*p* = 0.69		NasT vs. NasNT	9.56 ± 1.44 vs. 8.24 ± 0.79	*p* = 0.13
**Layer**	**Region**	**Mean ± SD**	**Statistical Significance**	**Layer**	**Region**	**Mean ± SD**	**Statistical Significance**
LVLS	D vs. V	27.42 ± 6.28 vs. 7.99 ± 0.70	*p* < 0.001	WCT	D vs. V	27.42 ± 6.28 vs. 7.99 ± 0.70	*p* < 0.001
	D vs. TempT	27.42 ± 6.28 vs. 10.84 ± 2.24	*p* = 1,00		D vs. TempT	27.42 ± 6.28 vs. 10.84 ± 2.24	*p* = 0.99
	D vs. TempNT	27.42 ± 6.28 vs. 8.21 ± 0.75	*p* < 0.001		D vs. TempNT	27.42 ± 6.28 vs. 8.21 ± 0.75	*p* < 0.001
	D vs. NasT	27.42 ± 6.28 vs. 9.56 ± 1.44	*p* < 0.001		D vs. NasT	27.42 ± 6.28 vs. 9.56 ± 1.44	*p* < 0.001
	D vs. NasNT	27.42 ± 6.28 vs. 8.24 ± 0.79	*p* < 0.001		D vs. NasNT	27.42 ± 6.28 vs. 8.24 ± 0.79	*p* < 0.001
	V vs. TempT	7.99 ± 0.70 vs. 10.84 ± 2.24	*p* < 0.001		V vs. TempT	7.99 ± 0.70 vs. 10.84 ± 2.24	*p* < 0.001
	V vs. TempNT	7.99 ± 0.70 vs. 8.21 ± 0.75	*p* < 0.001		V vs. TempNT	7.99 ± 0.70 vs. 8.21 ± 0.75	*p* < 0.001
	V vs. NasT	7.99 ± 0.70 vs. 9.56 ± 1.44	*p* < 0.001		V vs. NasT	7.99 ± 0.70 vs. 9.56 ± 1.44	*p* < 0.001
	V vs. NasNT	7.99 ± 0.70 vs. 8.24 ± 0.79	*p* < 0.05		V vs. NasNT	7.99 ± 0.70 vs. 8.24 ± 0.79	*p* < 0.005
	TempT vs. TempNT	10.84 ± 2.24 vs. 8.21 ± 0.75	*p* < 0.001		TempT vs. TempNT	10.84 ± 2.24 vs. 8.21 ± 0.75	*p* < 0.001
	TempT vs. NasT	10.84 ± 2.24 vs. 9.56 ± 1.44	*p* < 0.001		TempT vs. NasT	10.84 ± 2.24 vs. 9.56 ± 1.44	*p* < 0.001
	TempT vs. NasNT	10.84 ± 2.24 vs. 8.24 ± 0.79	*p* < 0.001		TempT vs. NasNT	10.84 ± 2.24 vs. 8.24 ± 0.79	*p* < 0.001
	TempNT vs. NasT	8.21 ± 0.75 vs. 9.56 ± 1.44	*p* = 0.96		TempNT vs. NasT	8.21 ± 0.75 vs. 9.56 ± 1.44	*p* = 0.73
	TempNT vs. NasNT	8.21 ± 0.75 vs. 8.24 ± 0.79	*p* = 0.90		TempNT vs. NasNT	8.21 ± 0.75 vs. 8.24 ± 0.79	*p* = 0.85
	NasT vs. NasNT	9.56 ± 1.44 vs. 8.24 ± 0.79	*p* = 0.47		NasT vs. NasNT	9.56 ± 1.44 vs. 8.24 ± 0.79	*p* = 0.13

**Table 3 ijerph-20-03121-t003:** Ratio of the medium-sized vessel layer (2) thickness to the large vessel layer with lamina suprachoroidea (3) thickness in middle-aged (MA) and senior (SN) dogs in dorsal (D), ventral (V), temporal tapetal (TempT), temporal nontapetal (TempNT), nasal tapetal (NasT), and nasal nontapetal (NasNT) regions.

	MA	Mean ± SD	SN	Mean ± SD
D2:D3	0.28	32.59 ± 7.83:121.72 ± 24.58	0.26	31.62 ± 7.62:124.87 ± 28.32
V2:V3	0.32	24.43 ± 8.07:78.21 ± 19.22	0.33	24.33 ± 6.21:75.39 ± 14.28
TempT2:TempT3	0.32	38.15 ± 7.64:126.78 ± 35.78	0.37	45.43 ± 10.29:125.67 ± 19.96
TempNT2:TempNT3	0.32	27.11 ± 9.87:84.45 ± 17.24	0.32	30.86 ± 6.94:99.59 ± 22.98
NasT2:NasNT3	0.35	32.90 ± 10.02:96.78 ± 12.56	0.34	33.03 ± 5.40:102.33 ± 24.48
NasNT2:NasNT3	0.29	26.34 ± 8.90:89.72 ± 14.19	0.33	29.74 ± 7.50:90.59 ± 12.43

## Data Availability

Data are available upon reasonable request.

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
