# Peer review of "Morphological and Morphometric Analysis of Canine Choroidal Layers Using Spectral Domain Optical Coherence Tomography"

_ijerph, 2023, doi:10.3390/ijerph20043121_

Round 1

Reviewer 1 Report

Zwolska and coauthors present morphometric analysis of choroidal layers using spectral domain optical coherence tomography (OCT) in healthy dogs. Their results revealed that the retinal pigment epithelium-Bruch’s membrane-choriocapillaris complex (RPE-BmCc) was significantly thicker in the dorsal region and thinner in the V region than those in the other regions. The choroidal thickness profiles showed no significant differences between the age groups including the medium-size vessel layer (MSVL)-to- large vessel layer with lamina suprachoroidea (LVLS) thickness ratio. This study provides a reference value of the in vivo choroidal thickness in healthy canine eyes. There is limited OCT data available on the measurement of the choroidal thickness since most papers are focused on the retina. For this reason, this study is worth publishing. However, the followings need to be improved.

My specific comments are as follows:

1. Choroidal thickness is affected by blood pressure and cardiovascular diseases. The systemic blood pressure and cardiovascular disease status of the patients should be included.    

2. The study design of this manuscript is very similar to the author's previous paper (Zwolska et al., Acta Vet. Hung. 2021). Have the patients or OCT images in this manuscript ever been used in the previous paper? If so, it should be clearly specified.

3. In Figures 1 and 2: Are the images tapetal fundus or non-tapetal fundus? It should be noted in the figures. I recommend that the authors include images of both the tapetal and non-tapetal areas in Figure 1.

4. Lines 172-183: this part is written complicated and needs to be improved for better understanding. It apparently used horizontal scans only. Why did you not apply vertical scans to measure an area of the same distance in the optic nerve head for all four directions?   

5. Why the indicated areas were measured? Wouldn’t the peripapillary region and the area centralis are more important considering the clinical relevance?

6. The OCT and fundus images in Figure 3 have no optic disc. How was the exact distance from the optic disc measured?

7. Lines 79-80, “In our previous study, we analyzed whole choroidal thickness in dogs in the dorsal, ventral, temporal, and nasal regions.”: please add a reference.

8. Please add all statistic results in Table 2. The authors should give all statistical results even though they showed no significant differences.

9. In tables 2 and 3: Please add standard deviation or standard error with the mean value.

10. The RPE-BmCc of the tapetal fundus was only ~27 um in this study. It is much thinner than other reference values that the authors cited in this manuscript. Why do you think the RPE-BmCc thickness was thinner in your study?

11. Choroidal thickness of the Tt region was significantly thicker than those of other areas in your results due to the thicker MSVL and LVLS, but not RPE-BmCc-tapetum, in the Tt area. Why the vascular layers were thick in the Tt region? I recommend adding a discussion to this.

12. In this study, the standard deviation of the RPE-BmCc in the dorsal region is much higher than those in the ventral region. It is consistent that the thickness of the tapetum in OCT varies depending on the breed. Was there any detailed info for the included mixed-breed dogs?

13. Few paragraphs of the discussion are more likely a repeat of results in this manuscript. I would recommend adding more details about the clinical relevance of this study in the last paragraph of the discussion.

14. Please make it consistent for using abbreviations throughout the manuscript.  

Author Response

Thank you for your insightful review, which allowed us to improve the quality of our work. Please see the attachment with our responses.

Reviewer 2 Report

The comments are attached in the Pdf document

Author Response

(The authors gave the same response as above.)

Round 2

Reviewer 1 Report

Zwolska and coauthors present a morphometric analysis of choroidal layers using spectral domain optical coherence tomography (OCT) in healthy dogs. Reference values of the in vivo The choroidal thickness of the healthy in vivo canine eyes provided in this study is important and essential information for other researchers and clinicians. This manuscript has been revised well by reflecting the reviewer’s comments, and the results are much more precise with the revised tables. There are some minor errors in the manuscript that need to be corrected (e.g. inconsistency for the italic form of “P”; a capital letter in the middle of the sentence; different font sizes in line 174; errors for the reference format; tables with line changes in inappropriate positions). 

Author Response

Thank you very much for the rewiev and for your time.

We have corrected the minor errors we had found - italic form of “P” was modified, font size in line 174 was changed, reference management in the manuscript and in the “References” section were changed, tables were modified.

Reviewer 2 Report

Thanks to have review the paper after my advices and comments . 

Author Response

Thank you very much for the rewiev and for your time.

Round 3

Reviewer 1 Report

Zwolska and coauthors present a morphometric analysis of choroidal layers using spectral domain optical coherence tomography (OCT) in healthy dogs. Reference values of the choroidal thickness in healthy canine eyes were provided in this study. It is important information for other researchers and clinicians. Minor errors previously pointed out have been corrected in the revised manuscript.